# Effectiveness of Personalized Low Salicylate Diet in the Management of Salicylates Hypersensitive Patients: Interventional Study

**DOI:** 10.3390/nu13030991

**Published:** 2021-03-19

**Authors:** Paulina K. Kęszycka, Ewa Lange, Danuta Gajewska

**Affiliations:** Department of Dietetics, Institute of Human Nutrition Sciences, Warsaw University of Life Sciences (WULS), 159C Nowoursynowska Str., 02-776 Warsaw, Poland; ewa_lange@sggw.edu.pl

**Keywords:** dietary salicylates, low salicylate diet, NERD, salicylate hypersensitivity

## Abstract

Salicylic acid and its derivatives (including acetylsalicylic acid/aspirin) are popular in medicine. They also occur naturally in many food products. The aim of the study was to investigate the effect of the personalized low salicylate diet (PLSD) on the reduction of asthma, rhinosinusitis and urticaria symptoms in patients with hypersensitivity to aspirin (ASA) or nonsteroidal anti-inflammatory drugs (NSAIDs). To achieve the research goal, a prospective, nonrandomized, baseline-controlled intervention study was conducted. Thirty patients diagnosed with NSAIDs hypersensitivity, who despite pharmacotherapy had symptoms of hypersensitivity, were included in the study. The PLSD was recommended for all participants for a period of two to four weeks. The intensity of subjectively declared symptoms of asthma, rhinosinusitis and urticaria were measured before and after dietary intervention, using, respectively, the asthma control test (ACT), the sino-nasal outcome test (SNOT-22) and the four-item itch questionnaire (FIIQ). Diet adherence and salicylate intake were measured by a 3-day food record. The severity of symptoms improved significantly after the intervention. The median of the ACT score was 24 scores before and 25 after the dietary intervention (*p* < 0.002), the median of the SNOT-22 score was 25 before and 13 after a dietary intervention (*p* < 0.0002) and the median of the FIIQ score was 5 before and 0 after a dietary intervention (*p* < 0.0002). The intake of salicylates decreased from 0.79 mg/day (before intervention) to 0.15 mg/day (*p* < 0.001) (during intervention). Although the usefulness of a low salicylate diet in the treatment of salicylate hypersensitivity is controversial, the results of our study indicate that the PLSD may have a positive effect in reducing symptoms of salicylate hypersensitivity and could be an additional tool supporting the therapy of these patients.

## 1. Introduction

Salicylates (C6H4(OH)COO-R) are bioactive organic compounds naturally occurring in food products. The most popular derivative of salicylic acid is acetylsalicylic acid. The latter is frequently used in medicine as an analgesic, antipyretic and anti-inflammatory agent. With long-term use, it also presents anticoagulant properties [1,2,3,4].

Studies confirm that salicylates (including acetylsalicylic acid (ASA)) occur naturally in foods [5,6]. The presence of salicylates in plants is physiologically justified [7,8], and their amount in plants may depend on different factors, including the morphological part of the plant, species, or cultivation method. This may explain the difference in the determination of the salicylates content in food (dietary salicylates) between studies of different authors [1,2,3,4,9,10,11,12]. Foods high in salicylates include legumes (e.g., lentils, beans), vegetables (e.g., cauliflowers, pickled vegetables), fruits (e.g., strawberries, plums, watermelons, raspberries), some cereals (e.g., buckwheat, oat or corn), herbs and spices [1,4,11,12].

The health-promoting properties of acetylsalicylic acid and other salicylates (e.g., sodium salicylate, salsalate, sulfasalazine) administered in therapeutic doses are well-known [2,5]. However, the effects of salicylates consumed with food are often overlooked, and their role underestimated. Some studies show that even a small amount of salicylates provided by food can have positive effects on human health [13,14,15,16,17]. Moreover, the level of salicylate metabolites (salicyluric acid) found in the urine of people on a plant-based diet can be compared with the amount determined in the urine of people taking cardioprotective doses of acetylsalicylic acid [18].

However, for some individuals, the intake of acetylsalicylic acid (popular aspirin) and other nonsteroidal anti-inflammatory drugs (NSAIDs) can cause hypersensitivity reactions. Taking into account that aspirin is one of the most commonly used anti-inflammatory drugs, this hypersensitivity is frequently called aspirin hypersensitivity. Symptoms of this hypersensitivity can also occur after an administration of other cyclooxygenase inhibitors, i.e., other NSAIDs. Therefore, it has been suggested that in this type of hypersensitivity, the term “aspirin” be replaced with “NSAID” [19].

Aspirin and NSAIDs can exacerbate asthma, rhinosinusitis, and urticaria [5,20,21,22,23,24]. Yet, the incidence of NSAIDs hypersensitivity and, to a greater degree, dietary salicylates hypersensitivity is poorly known. NSAIDs-exacerbated respiratory disease (NERD) is probably one of the most frequently described in the literature. In the European population, the mean prevalence of NSAID-induced dyspnea was 1.9% [25]. Aspirin-exacerbated respiratory disease (AERD) is noted in 7% of patients with asthma [26]. It is also accepted that in approximately 20–30% of patients with chronic urticaria, aspirin may exacerbate urticaria symptoms [27]. The Cavkaytar single-blind, placebo-controlled provocation tests with aspirin showed hypersensitivity to aspirin in 24% of children with chronic persistent urticaria and in 1 out of 10 patients with chronic recurrent urticaria [28]. The question remains whether the intake of dietary salicylates can exacerbate symptoms in patients with hypersensitivity to ASA and NSAIDs and whether a low salicylate diet is justified in the treatment of NSAIDs hypersensitivity.

Position papers developed by the European Academy of Allergy and Clinical Immunology do not recommend the use of salicylate-free diet as supportive therapy for patients with the NSAIDs-exacerbated respiratory disease (NERD) [19]. The grade of recommendation is marked as D, meaning that there is a lack of research in this area. Thus, it seems that studies on the effect of a low salicylate diet on the efficacy of therapy for patients with NSAID hypersensitivity are needed. Indeed, experts’ doubts about the efficacy of a low salicylates diet may result from conflicting and outdated dietary recommendations and the small number of studies in this area. Nonetheless, seeing the influence the diet can have on the patients’ quality of life and on the reduction of symptoms caused by hypersensitivity to salicylates [5,29,30], it was decided to examine the effectiveness of an elimination diet based on up-to-date data concerning salicylates content in food [4,11,31,32].

The aim of the study was to investigate the effectiveness of the personalized low salicylate diet (or the “PLSD”) on the reduction of asthma, rhinosinusitis and urticaria symptoms among patients with hypersensitivity to aspirin (ASA) or nonsteroidal anti-inflammatory drugs (NSAIDs). Dietary principles were developed on the basis of recent databases of salicylates content in food.

## 2. Materials and Methods

### 2.1. Study Design

A prospective, nonrandomized, baseline-controlled intervention study was conducted to achieve the research goal. We have decided to use a model in which patients’ health status on the PLSD was compared with status before therapy. For ethical reasons, the crossover study model was abandoned. In fact, switching from a low salicylates diet to a high salicylates diet could result in a sudden increase in salicylates supply and be dangerous for the patients. Furthermore, the heterogeneity of the group in relation to hypersensitivity symptoms was not conducive to the formation of a control group. A detailed scheme of the study is shown in Figure 1.

The study was conducted at the Department of Dietetics at the Warsaw University of Life Sciences-SGGW between 2017 and 2019. The study model was approved by the Ethics Committee of WULS-SGGW and obtained its consent (No 06p/2017). All procedures were conducted according to the guidelines provided in the Declaration of Helsinki, while participants gave their written informed consent to participate in the study.

### 2.2. Study Participants

Thirty-four subjects (31 women and 3 men) meeting the following criteria were included in the study:Age > 18 years old;A medical diagnosis of hypersensitivity to NSAIDs confirmed by an allergist;Persistent self-reported symptoms, typical of salicylates hypersensitivity (rhinitis and/or asthma, and/or urticaria), despite pharmacological treatment and NSAIDs avoidance;Intake of a stable dose of medication for at least 2 months prior to the commencement.

Exclusion criteria were as follows: pregnancy, serious diet-related diseases requiring additional significant dietary modifications, such as diabetes, chronic kidney disease, celiac disease, liver disease, intestinal diseases, cancer, a history of immunodeficiency or anaphylaxis. All participants were referred to the study by allergists.

Two women and 2 men did not complete the study as they found the PLSD too difficult to follow. Therefore, all results are presented for 30 subjects (29 women and 1 man). Pharmacological treatment of the patients was not modified throughout the study.

### 2.3. Dietary Intervention

The dietary intervention consisted of the implementation of the PLSD by the patients for 2 to 4 weeks. This was an elimination diet that consisted of avoiding food products rich in salicylates. Twenty-three subjects followed the PLSD for 2 weeks. For the remaining 7 subjects, the diet was extended for an additional 2 weeks because of noncompliance with the diet. The assumptions of the low salicylate diet were based on the available literature [1,2,3,4,9,10] and on our own research results [11,32].

Each patient was given a list of food products divided into two categories:Salicylate-free products, safe for consumption, such as millet, barley, wheat, poultry, fish, eggs, milk, butter, lard, peeled pears or some varieties of apples;Containing salicylates, which should be limited to a certain amount per day.

Foods containing salicylates were categorized as low, medium, high or very high in salicylates content Products containing a low amount of salicylates were allowed in the PLSD (e.g., bananas, chives, apples, lemon, cabbage, carrot, pumpkin, onion, olive oil, cheese, yogurt, beef and pork). Products rich in salicylates were not allowed (e.g., herbs, spices, strawberries, grapes, plums, canned vegetables, pickles, cauliflower or processed foods, such as sausages, sauces and ready-to-eat meals).

The PLSD was based mainly on salicylate-free products. Low salicylate products were limited to 5 servings/day and no more than 1 serving/meal. These products contain negligible amounts of salicylates. According to the database of Kęszycka et al. [11], they contain less than 0.05 mg of total salicylates/serving. Products with higher salicylates content (described as medium, high and very high salicylates) were not recommended. It was estimated that the total supply of salicylates in the PLSD should not exceed 0.25 mg per day.

Each patient received individual recommendations related to the total number of servings of food that was either salicylate-free or containing salicylates, according to the patient’s total energy requirement. In order to facilitate the task of planning a balanced diet for the patients, a system of food exchangers was prepared. The patients could choose food products of high nutritional value and low in salicylates from all food groups. This allowed tailoring a diet plan meeting the individual needs of the patients and providing adequate energy, macronutrients and micronutrient amounts. The patients were given sample menus and recipes for low salicylates meals.

The energy requirements for the patients were calculated using the Mifflin-St Jeor formula [33]. Physical activity declared by the patients was taken into account. The recommended diet was based on unprocessed or minimally processed food, free of preservatives, dyes and other artificial food additives.

All patients were under the healthcare of the dietitian (P.K.K.), who specialized in the management of food intolerance and allergies. Twenty-three patients followed 3 in-person visits: (1) an initial visit (lasting about 90 min), (2) an educational visit prior to the dietary intervention (90 min), (3) a final visit after 2 weeks following the PLSD (60 min). The remaining 7 patients followed 4 in-person visits: (1) an initial visit (lasting about 90 min), (2) an educational visit before the dietary intervention (90 min), (3) a second educational visit after 2 weeks of following the PLSD (60 min), (4) a final visit after another 2 weeks of following the PLSD (60 min).

For the whole duration of the study, all patients were in phone contact with the dietitian.

### 2.4. Questionnaires

Diet adherence, salicylate intake and the number of servings of salicylate-containing products were assessed before the intervention and during the last 3 days of the intervention, using a 3-day food record.

Patients’ health status and nutritional status were assessed during the first/initial visit and the final visit, using a standard questionnaire, which is routinely used in dietary interviews. This questionnaire contained open questions about medical history and health status, medical test results, medications and supplements, nutrition and lifestyle.

Symptoms associated with hypersensitivity to NSAIDs were assessed using subjective symptom questionnaires:The asthma control test (ACT) was used to assess asthma symptoms [34]. The test was conducted before the dietary intervention at the educational visit and on the last day of the PLSD use during the final visit. Higher scores indicated better asthma control;The sino-nasal outcome test (SNOT-22) was used to assess the symptoms of rhinosinusitis [35]. The test was conducted before the intervention, during the educational visit and on the last day of the PLSD use, during the final visit. Higher scores indicated greater severity of rhinosinusitis symptoms;The food-item itch questionnaire (FIIQ) was used to assess pruritus induced by urticaria [36]. The test was performed 3 times before the dietary intervention, and 3 times during the last 3 days of the intervention. The results obtained were presented as the mean of 3 repetitions. The results are presented as the mean of three replications. Higher scores indicated a higher severity of pruritus.

### 2.5. Assessment of Salicylates Intake

Diet adherence was assessed by comparing the number of servings of salicylate-containing products consumed by patients with the recommended number of servings. Data were obtained from a 3-day food record. To calculate total salicylates intake, we have used several databases on salicylates content in food, including the own studies [11,31,32] as well as Malakar et al.’s study [4].

### 2.6. Statistical Analysis

All data were collected and entered into an Excel database (Microsoft Corp., Redmont, WA, USA). Data were analyzed using Statistica 14 (StatSoft Inc., Tulsa, OK, USA). The normality of data distribution was verified using the Shapiro–Wilk test. Due to the nonparametric distribution of all obtained results, the Wilcoxon signed-rank test was used to assess the significance of differences between the results. Correlations were checked using Spearman’s rank correlation coefficient. The *p* < 0.05 was accepted in order to verify significance.

## 3. Results

The characteristics of the study group are shown in Table 1. At baseline, 3 subjects (10%) had a BMI < 18.5 kg/m^2^ indicating a low body mass, 17 subjects (57%) had a normal body weight, while 6 subjects (20%) were overweight, and 4 subjects (13%) were obese. The nutritional status of patients did not significantly change during the dietary intervention, as the recommended PLSD was not a calorie-restricted diet.

Despite pharmacotherapy, all patients declared having symptoms of hypersensitivity. Thirteen patients (43%) declared having all symptoms associated with hypersensitivity (asthma, rhinosinusitis, and urticaria). The most common symptom of hypersensitivity, declared by 27 out of 30 patients, was rhinosinusitis. The majority of patients were treated with antihistamines (83%) and glucocorticosteroids (70%). Table 2 presents the group of medication used by patients during the study.

Before the dietary intervention, more than half of the patients (57%) declared following a low salicylate diet, based on information obtained via the Internet or from their physicians. However, the assessment by a dietitian of the patients’ diets leads to different findings. Only 3 patients (10%) were, in fact, following a low salicylate diet before the dietary intervention. The diets of 14 patients (47%) were found to be not low in salicylates, despite what the patients believed. Baseline analysis of patients’ diet revealed that 26 subjects (87%) consumed 4 or more servings of salicylate-rich foods per day. Three individuals (10%) consumed 1–2 servings of salicylate-rich foods per day. One person, based on self-observation and physician recommendations prior to the dietary intervention, consumed only 1–3 servings of salicylate-rich foods per week.

During the dietary intervention, 22 subjects (60%) consumed 1–3 servings of salicylate-rich products per day, and 6 individuals (20%) consumed less than 3 servings of salicylate-containing products per week. Only 2 patients (7%) consumed 4 or more servings of salicylate-rich products per day.

The assessment of diet adherence by the dietitian showed that 20 subjects (67%) adhered partially or completely to the dietary recommendations. However, there was no significant correlation between closer patients’ adherence to the PLSD (as assessed by the dietitian) and greater reduction in self-reported symptoms of hypersensitivity.

In comparison with the beginning of the study, patients consumed significantly fewer salicylates after following the PLSD for the recommended period. Salicylates intake estimated on the basis of the database of Kęszycka et al. [11] and Gajewska et al. [32] is presented in Table 3. We further confirmed the aforementioned differences in salicylate consumption by calculating the dietary salicylate intake according to the Malakar et al. database [4]. The median salicylates intake was 4.36 mg/day and 1.98 mg/day, before and after the dietary intervention, respectively (*p* < 0.01).

The following of the PLSD diet resulted in significant improvement in hypersensitivity symptoms among 22 patients (73.3%) and a moderate improvement in 4 patients (13.3%), while 4 patients (13.3%) showed no improvement and even worsening of some parameters.

Table 4 shows the results of self-reported symptoms, measured by the ACT, the SNOT-22, and the FIIQ tests. Of the 20 patients diagnosed with asthma, 14 patients (70.00%) had improved asthma control after the PLSD, 5 patients (25.00%) had unchanged symptom severity, and 1 patient had worsened asthma control after the PLSD.

In 77.78% of patients diagnosed with rhinosinusitis, a lower SNOT-22 score was obtained after the PLSD, indicating a reduction in symptom severity. No change in symptom severity was noted for 14.82% of patients, while 2 patients (7.41%) had an increase in symptoms.

After the PLSD application, 20 patients (86.96% of patients with diagnosed urticaria) declared a remission of symptoms of urticaria. In 2 patients (8.70%), the severity of symptoms did not change, while 1 patient declared an increase in perceived pruritus. Individual changes in test results and salicylates intake before and after the dietary intervention are demonstrated in Figure 2.

We observed a significant correlation between changes in salicylate intake and changes in the reduction of symptoms of rhinosinusitis assessed by the SNOT-22 questionnaire (r = 0.89, *p* < 0.05). A lower salicylate intake was associated with a greater reduction in symptoms. An important correlation was also found between the salicylates intake and the FIIQ score. With higher intakes of salicylates we observed higher FIIQ scores (r > 0.97, *p* < 0.05). However, this observation was statistically significant only when the dietary salicylates intake was calculated using Malakar et al.’s database [4].

## 4. Discussion

The mechanisms responsible for hypersensitivity reactions to aspirin and NSAIDs are increasingly explained and described in greater detail [19]. However, the effect of dietary salicylates on COX-1 and COX-2 is not fully understood [30]. Routine recommendation of a low salicylate diet for patients with hypersensitivity to NSAIDs continues to be controversial. This stems from a lack of good understanding of the mechanisms responsible for hypersensitivity to food-derived salicylates, the small number of studies investigating salicylates content in food, and the lack of unambiguous, validated dietary recommendations for patients suffering from salicylates hypersensitivity.

Hypersensitivity to NSAIDs is defined as type B adverse drug reaction, i.e., unpredictable, occurring only in susceptible individuals [37]. It can be divided into immune-mediated hypersensitivity and nonallergic hypersensitivity [37]. Therefore, the symptoms caused by this hypersensitivity may be diverse: immediate or delayed, affecting the skin, the respiratory tract or other organs, and may vary from mild to severe (such as anaphylaxis or even death) [5,37]. The most common symptoms of NSAIDs hypersensitivity are aspirin triad described by Samter, characterized by the presence of nasal polyposis, asthma and NSAIDs intolerance [22]. In addition, urticaria, angioedema, rhinosinusitis and other nonspecific symptoms may also occur [5,37,38,39].

A certain group of patients develops hypersensitivity symptoms despite excluding NSAIDs and using appropriate pharmacotherapy [5,29,30]. Therapeutic solutions for these patients are, therefore, required. One of the solutions is desensitization. However, it does not bring the expected results in all patients, it is expensive, and the side effects thereof lead up to 30% of patients undergoing desensitization to NSAIDs to resign from completing the procedure [40].

Because researchers, physicians, and NERD patients themselves often recognize the impact of diet on symptom severity [5,29,30,41], it seems advisable to look for a diet that could support therapy for hypersensitivity to NSAIDs. Levy et al. 2016 suggested that the low-cost associated with the introduction of a low salicylate diet, the simplicity of its introduction compared to other medical procedures, and the potential benefits of its use outweigh the possible risks. They also believe that the introduction of such a diet may be considered as an adjunctive treatment option in patients with AERD and sinonasal symptoms not controlled by standard treatment [42]. As in the case of other food allergies and intolerances, a diet eliminating salicylates seems to be useful, at least for diagnostic reasons.

Effective and safe diet therapy is an individualized diet that cannot be unified. It requires an individual approach to the patients, their needs, and observation of symptoms associated with the consumption of different products. This is particularly important in the case of salicylates hypersensitivity, as there are many uncertainties and conflicting results about the salicylate content in foods [1,2,3,4,9,10,11,12].

It is likely that these ambiguities contributed to the opinion described in the Statement by the Food Allergy Working Group of the German Society for Allergology and Clinical Immunology in 2020. The Working Group concluded that reducing salicylate intake in the treatment of ASA hypersensitivity has no pathophysiological basis and may cause risks of nutritional deficiencies. Therefore, this type of diet is not recommended [43].

Doubts about the effect of salicylates from food on patients’ health are further supported by the fact that the salicylates content of food is extremely low compared to the number of salicylates used in therapeutic doses. This position was confirmed by Janssen et al. in their paper from 1996, in which they estimated salicylates intake at 0–6 mg/day and concluded that such a small amount is probably too low to influence disease risk or behavior in children [44]. Unfortunately, the authors did not refer to food hypersensitivities, in which often trace amounts of pseudo-allergens suffice to cause the symptoms.

The argument confirming the validity of the low salicylate diet is also supported by studies showing its positive health effects in patients with hypersensitivity to NSAIDs. Despite the lack of a proven mechanism, various teams of researchers have recognized its positive effects on the patient’s health [5,29,30,45,46]. In contrast, Philpott et al., in their work, note that patients with chronic rhinosinusitis reported exacerbation of symptoms most commonly after consuming products potentially rich in salicylates [41].

Conflicting experiences and doubts about the effectiveness of the low salicylate diet are probably also due to the fact that the tools and databases of salicylates content in foods for diet planning are very limited. The “aspirin diet” that patients find on the Internet is often based on outdated studies of salicylates content in foods and, therefore, may not be effective for patients with hypersensitivity to salicylates. Additionally, patients seeking help on the Internet are rarely informed that a restrictive low salicylate diet can be followed only for a short period of time to diagnose or suppress symptoms when pharmacotherapy is not sufficient. The continuation of a highly restrictive diet can be detrimental to human health. To be safe for patients, the diet should always be introduced and monitored by an experienced dietitian. It should be recommended for a period not exceeding four weeks because of the possibility of nutritional deficiencies. The gradual expansion of the diet, according to the patient’s individual salicylates tolerance, seems necessary.

The results of our study revealed the positive effect of the PLSD on reducing symptoms characteristic of NSAIDs hypersensitivity, which in turn may confirm that hypersensitivity to dietary salicylates does occur. However, as hypersensitivity symptoms did not resolve in all patients, we cannot unequivocally exclude that factors other than dietary salicylates may cause hypersensitivity to NSAIDs.

The aim of this study was to investigate the effectiveness of the PLSD on the reduction of salicylate hypersensitivity symptoms. Our recommendations focused on lowering salicylate intake from food products. However, we cannot exclude that this dietary counseling positively influenced the general dietary behavior of the patients. Perhaps patients paid more attention to their dietary choices. Yet, we obtained satisfying results as in similar studies, which were focused on the effects of salicylate intake on hypersensitivity symptoms [5,29,30].

We observed that patients on the PLSD in our study reported that their asthma was better controlled. They also stated that the extent, severity, frequency, and sleep disturbance associated with urticaria were less significant, and the symptoms of rhinosinusitis were less severe after introducing the PLSD. The reduction of symptoms was particularly observed in cases of the need to blow nose, nasal blockage, sneezing, runny nose and post-nasal discharge. Sommer et al. obtained similar results in their study. In addition to subjective self-reported tests, they used an objective scale to assess the endoscopic findings. They also found that the low-salicylate diet may offer a novel treatment adjunct to the current management of AERD [30]. The diet should be based on up-to-date databases of salicylates content in foods and detailed dietary history of the patient. The dietitian should monitor and modify the patient’s diet to reduce the risk of vitamin and mineral deficiencies. Particular attention should be paid to limiting the intake of products previously considered low in salicylates, but, on the basis of recent studies, classified as salicylate-rich (e.g., oats, buckwheat, cauliflower, rice).

The PLSD model that was used in our study can be safely recommended for patients with hypersensitivity to NSAIDs for whom previous therapy was inefficient. The PLSD applied in the study provided a minimal amount of salicylates. The latter were derived from foods essential in a well-balanced diet. Moreover, this, in fact, the restrictive diet was followed only for 2–4 weeks. Such a period of time is sufficient to alleviate the hypersensitivity symptoms and confirm or exclude the influence of dietary salicylates on the patient’s symptoms. It seems then advisable to gradually introduce foods containing small to medium amounts of salicylates and to establish individual safe doses of dietary salicylates in patients who responded positively to the dietary intervention [5]. This approach would help to avoid recurrence of symptoms and to balance the diet properly. Among patients for whom a low salicylate diet has not brought the desired results, it is worth looking for other causes of health problems.

Our findings also show that more than half of the patients (56.67%) rated their diet as low in salicylates before the dietary intervention, while in fact, only 3 patients (10.00%) actually followed a low salicylate diet. Thus, it is possible that much of the controversy regarding the effectiveness of this diet is caused by inaccurate dietary recommendations given to patients. It is worth adding that one salicylate-rich product incorrectly recommended in a low salicylate diet suffices to exacerbate symptoms of salicylates hypersensitivity.

Similar studies in which symptoms of hypersensitivity to salicylates were significantly reduced after the introduction of a low salicylate diet can be found in the literature [5,29,30]. The key issue seems to be the individual approach to the patient, taking into account not only the salicylates content of the diet but also other potential factors exacerbating hypersensitivity symptoms.

The planning of an appropriate low salicylate diet can be difficult, as the salicylates content is highly variable and depends on many factors, such as the plant species and the way the plants are grown [1,11]. Organic cultivation may result in higher amounts of salicylates in plants, as salicylic acid is a phytohormone that protects plants from pathogens [47]. The degree of ripeness of vegetables and fruits, as well as the presence of substances added to food, are also important factors [5].

In addition, given the widespread use of salicylates in the food industry, it seems necessary to recommend that patients consume minimally processed food products coming from known, reliable suppliers. Additives may also constitute a risk for hypersensitive patients. Benzoic acid is known to be a substrate for salicylic acid [48].

Certain study limitations must be considered. The sample size was relatively small. The trial was not blinded, as this was not possible for a study of this nature. The diet was quite restrictive, which made it difficult for some patients to adhere to it. The tests used to assess the severity of hypersensitivity symptoms were subjective in nature. However, they were used to evaluate the severity of the symptoms, the subjective evaluation, which has a significant impact on the patients’ quality of life. It is worth emphasizing that the PLSD was tailored to the individual nutritional needs and preferences of the patients.

## 5. Conclusions

The personalized low salicylate diet may have a positive effect on reducing self-reported symptoms of asthma, rhinosinusitis, and urticaria, although it is not effective in all patients diagnosed with hypersensitivity to ASA or NSAIDs. The low salicylates diet may be a helpful new tool to support salicylates hypersensitivity therapy, helping to mitigate the symptoms and improve patient well-being. However, further research is needed on the salicylates content of foods, and thus, some modifications of the low salicylate diet may be necessary. Further research is also needed to understand the mechanism of the effect of salicylates in food on the development of food hypersensitivity symptoms.

## Figures and Tables

**Figure 1 nutrients-13-00991-f001:**
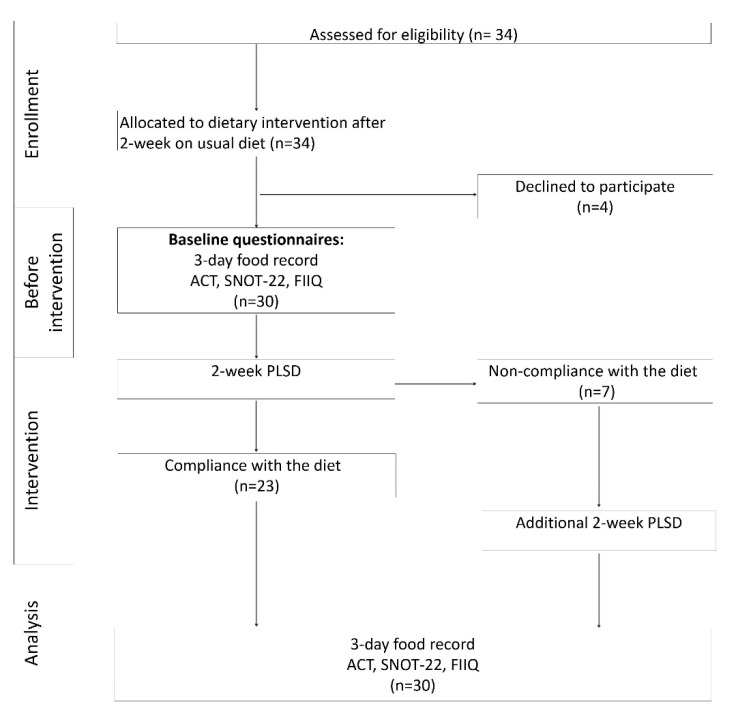
Study design. PLSD—personalized low salicylate diet, ACT—asthma control test, SNOT-22—sino-nasal outcome test, FIIQ—four-item itch questionnaire.

**Figure 2 nutrients-13-00991-f002:**
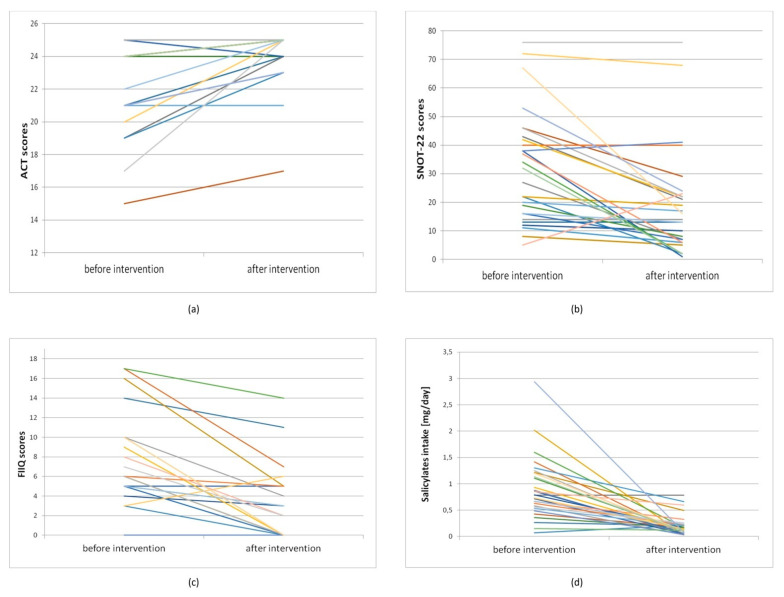
Individual results of the study participants: (**a**) individual results of “asthma control test” (ACT) scores before and after dietary intervention, *n* = 20; (**b**) individual results of “sino-nasal outcome test” (SNOT-22) scores before and after dietary intervention, *n* = 27; (**c**) individual results of “four-item itch questionnaire” (FIIQ) scores before and after dietary intervention, *n* = 23; (**d**) individual salicylates intake before and after dietary intervention, *n* = 30.

**Table 1 nutrients-13-00991-t001:** Baseline characteristics of the study group, *n* = 30.

	Mean	SD	Min–Max	Median
Age, years	40.33	16.29	18.00–83.00	37.50
Weight, kg	63.36	10.17	47.00–83.70	62.55
BMI, kg/m^2^	23.28	4.25	17.92–32.70	22.02
Disease duration, years	3.32	3.04	0.25–10.00	2.00
	Number of patients		Percent (%)	
Smokers	2		6.67	
Food allergies	10		33.33	
Inhalation allergies	4		13.33	
Contact allergies	2		6.67	
No allergies	14		46.67	
Family history of aspirin sensitivity	7		23.33	
Symptoms	Number of patients		Percent (%)	
Urticaria (only)	2		6.67	
Rhinosinusitis (only)	1		3.33	
Urticaria and asthma	1		3.33	
Urticaria and rhinosinusitis	7		23.33	
Asthma and rhinosinusitis	6		20.00	
Urticaria, asthma and rhinosinusitis	13		43.33	

SD—standard deviation, BMI—body mass index.

**Table 2 nutrients-13-00991-t002:** Group of medication used by study participants.

Group of Drugs	Number of Patients (*n* = 30)	Percent (%)
Antihistamines	25	83.33
Glucocorticosteroids	21	70.00
Leukotriene receptor antagonists	5	16.67
Hydroxyzine	2	6.67
Levothyroxine	6	20.00
Proton pump inhibitors	3	10.00
Beta-blockers	2	6.67
Statins	1	3.33
Anticoagulants ^1^	2	6.67

^1^ Acetylsalicylic acid was not declared as an anticoagulant by the study participants.

**Table 3 nutrients-13-00991-t003:** Intake of dietary salicylates before and after dietary intervention among study participants.

	Salicylate Intake (*n* = 30)			
	Baseline	PLSD	Statistical Significance ^a^	Δ PLSD vs. Baseline
Mean, mg/day	0.90	0.20		−0.70
Median, mg/day	0.79	0.15	*p* < 0.001	−0.60
Min, mg/day	0.07	0.03		−2.86
Max, mg/day	2.94	0.79		0.14
SD	0.58	0.19		0.61

PLSD—personalized low salicylate diet, ^a^—versus baseline, SD—standard deviation.

**Table 4 nutrients-13-00991-t004:** Comparison of the ACT scores, the SNOT-22 scores and the four-item itch questionnaire scores before and after dietary intervention.

Participants		Initial Test before Intervention	Retest after Intervention	*p* ^1^	∆ after vs. before Intervention
ACT score					
All *N* = 30	Mean	23.00	24.27		1.27
	Median	24.00	25.00	0.001332	0.00
	Min	15.00	17.00		−1
	Max	25.00	25.00		8
	SD	2.75	1.66		2.02
With asthma *N* = 20	Mean	22.00	23.90		1.90
	Median	23.00	25.00	0.001332	1.00
	Min	15.00	17.00		−1.00
	Max	25.00	25.00		8.00
	SD	2.90	1.94		2.22
SNOT-22 score					
All *N* = 30	Mean	28.97	17.13		−12.07
	Median	24.50	13.00	0.000175	−4.50
	Min	0	0		−51.00
	Max	76.00	76.00		11.00
	SD	20.89	18.61		14.73
With rhinosinusitis *N* = 27	Mean	32.19	19.04		−13.41
	Median	32.00	14.00	0.000175	−9.00
	Min	5.00	1.00		−51
	Max	76.00	76.00		11
	SD	19.47	18.67		14.94
FIIQ score					
All *N* = 30	Mean	5.90	2.43		−3.57
	Median	5.00	0.00	0.000152	−3.00
	Min	0	0		−11
	Max	17	14		3
	SD	5.11	3.51		3.75
With urticaria *N* = 23	Mean	7.70	3.04		−4.65
	Median	6.00	2.00	0.000152	−5.00
	Min	0	0		−11
	Max	17.00	14.00		3
	SD	4.60	3.78		3.72

^1^ verified using Wilcoxon signed-rank test; ACT—asthma control test, SNOT-22—sino-nasal outcome test, FIIQ—four-item itch questionnaire.

## Data Availability

The data presented in this study are available on request from the corresponding author. The data are not publicly available due to patients privacy restrictions.

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
