# Peer review of "Effectiveness of Personalized Low Salicylate Diet in the Management of Salicylates Hypersensitive Patients: Interventional Study"

_nutrients, 2021, doi:10.3390/nu13030991_

Round 1

Reviewer 1 Report

This manuscript explored to assess the effectiveness of low salicylates diet intervention for patients with hypersensitivity to NSAIDs. Although the role of dietary intake of salicylates on hypersensitivity is controversial, it is worthwhile and interesting study as authors attempted to address the issue by performing a diet intervention. As authors also mentioned for some, this study has limitations, too, such as small sample size and relying on the information heavily provided by participants on low salicylate diet, and ACT, SNOT-22, and FIIQ evaluations. In addition, in depth discussion of other possible confounding factors that may be accompanied by or associated with the restrictive low salicylate diet for 2 weeks (i.g., other beneficial ingredients in the diet or change of lifestyle) is necessary, since this study does not provide a direct evidence that links low salicylates to hypersensitivity relief. Lastly, I recommend to reorganize and rephrase the introduction and discussion sections to be logical and succinct to the topic.

Reviewer 2 Report

General comment: The research article entitled “Effectiveness of Personalized Low Salicylates Diet in the Man agement of Salicylates Hypersensitive Patients: Interventional Study” is an intervention study that presents the effect of a low salicylates diet on patients with hypersensitivity. This is a well-organized study, with sufficient methodology and adequate description of the results. Some minor corrections are required for the improvement of the manuscript.

Abstract: The Abstract is well written and adequately presents the aim and the basic results of the study.

-Line 11. Authors could add some words for the description of the study eg this intervention prospective study or similar.

-Lines 22-24. The decimals of pv could be diminished.  

Introduction: The introduction section adequately covers the need for investigation of a dier low on salicylates for specific uses.

-Line 37. Authors could refer examples of foods rich in salicylates.

Materials and Methods:  The materials and methods are adequately presented.

-Did authors use questionaries for the investigation of the dietary and medical history of the participants? Please add a sentence.

-What kind of questionnaire used for the foods rich or low on salicylates? Is this a validated questionnaire?

-Could authors add a sentence about the basic foods low or rich in salicylates that studied?

Results: The results of the study are analytically presented. Tables and Figures are adequate explain the findings of the study.

Discussion: The results of study are sufficiently discussed.

Conclusion: The conclusion is adequate and summarizes the main text.

Bibliography/References: The references used by the authors cover adequately the relative scientific field and the aims of the study.
